# Gaussian and Lerch Models for Unimodal Time Series Forcasting

**DOI:** 10.3390/e25101474

**Published:** 2023-10-22

**Authors:** Azzouz Dermoune, Daoud Ounaissi, Yousri Slaoui

**Affiliations:** 1CNRS, Laboratoire Paul Painlev, UMR 8524, Université de Lille, 59653 Villeneuve d’ascq, France; azzouz.dermoune@univ-lille.fr; 2Ecole Supérieure des Agricultures, 49000 Angers, France; d.ounaissi@groupe-esa.com; 3Laboratoire de Mathématiques et Applications, Université de Poitiers, 11 Boulevard Marie et Pierre Curie, 86073 Poitiers, France

**Keywords:** Gaussian model, Lerch model, least absolute deviation, daily infection, simplex algorithm, Nelder–Mead, optim function

## Abstract

We consider unimodal time series forecasting. We propose Gaussian and Lerch models for this forecasting problem. The Gaussian model depends on three parameters and the Lerch model depends on four parameters. We estimate the unknown parameters by minimizing the sum of the absolute values of the residuals. We solve these minimizations with and without a weighted median and we compare both approaches. As a numerical application, we consider the daily infections of COVID-19 in China using the Gaussian and Lerch models. We derive a confident interval for the daily infections from each local minima.

## 1. Introduction

The least absolute deviations (LAD) method of curve-fitting proposed [1] consists of fitting the data $\left(x_{i},y_{i}\right)$ to a function $f(x_{i},\theta )$, with i=1,…,n. The parameter $\theta \in R^{p}$ minimizes the sum of absolute deviations
$$\begin{array}{c} \sum_{i=1}^{n}\left|y_{i}-f\left(x_{i},\theta \right)\right|. \end{array}$$
According to [2], in the linear regression case $f\left(x_{i},\theta \right)=\sum_{j=1}^{p}x_{ij}\theta_{j}$, the minimization of the quantity
$$\begin{array}{c} \sum_{i=1}^{n}\left|y_{i}-\sum_{j=1}^{p}x_{ij}\theta_{j}\right| \end{array}$$
was suggested by Boscovitch (1757) (some asymptotic results are given in [3]), see also [4,5,6]. The latter objective function is convex with respect to the parameter θ. Hence, it has only one minima but may have many minimizers.

The linear regression case of LAD optimization is inherently more complex than the minimization of the sum of squares. The interest in the LAD method is associated with the development of robust methods. The LAD method is more resistant to the outliers in the data (see [7,8]).

The aim of our work is to analyze LAD minimization using a nonlinear regression motivated by the daily infections of COVID-19 in China during the first wave. We denote I(t) the observed number of infected persons at time t∈[1,T] with T≤60 (see Figure 1). The variable t=1,…,T represents day 1,…,T.

Justified by the sigmoidal nature of a pandemic, we propose two models: the Gaussian model (see [9])
$$\begin{array}{c} I_{Gauss}\left(t\right)=a\exp\left(-\frac{{\left(t-l\right)}^{2}}{s^{2}}\right) \end{array}$$
and the Lerch model
$$\begin{array}{c} I_{Lerch}\left(t\right)=a\frac{z^{t}}{{\left(v+t\right)}^{s}}, \end{array}$$
as a prediction of the observed number of infected persons I(t) at time *t*.

The Gaussian parameter θ=(a,l,s), with a,l,s denote, respectively, the peak, the location of the peak, and the width of the first wave of COVID-19.

The Lerch probability distribution on the non-negative integers is proportional to the function $\frac{z^{t}}{{\left(v+t\right)}^{s}}$ with t=0,1,…, the parameters z∈(0,1) and v>0. The Lerch probability distribution is strongly unimodal when s<−1 and v≥1. In this case, its mode is at $[\frac{1}{z^{\frac{1}{s}}-1}-v]+1$, where [·] signifies taking the integer part, see [10].

To estimate the three parameters θ=(a,l,s) (respectively, θ=(a,z,v,s)) based on the *T* observations, we consider LAD nonlinear regression
$$\begin{array}{c} f\left(T,\theta \right)=\frac{\sum_{t=1}^{T}\left|I\left(t\right)-I_{m}\left(t\right)\right|}{T}, \end{array}$$
with the subscript m=Gauss (respectively, m=Lerch). The Gaussian model was studied in our previous work [11].

As in our previous work [11], we propose to solve our proposed LAD regression using the simplex Nelder–Mead algorithm implemented by the optim function in R software. The Nelder–Mead algorithm [7,12,13,14] is able to optimize functions without derivatives. It is a simplex method for finding a local minima of a function, the most widely used direct search method for solving optimization problems, and is considered one of the most popular derivative-free nonlinear optimization algorithms.

The output of the optim function depends on the initialization and is in general not a minimizer of the objective function. Restarting the Nelder–Mead algorithm from the last solution obtained (and continuing to restart it until there is no further improvement) can only improve the final solution and the latter is in general a local minimizer. Here is the iteration of the optim function until the convergence:

## 2. Probabilistic Interpretation of LAD Regression

Let us assume that
$$\begin{array}{c} I\left(t\right)=I_{m}\left(t\right)+e\left(t\right), \end{array}$$
where the errors (e(t)) are i.i.d. with the common probability distribution
$$\begin{array}{c} \frac{1}{2\lambda }\exp\left(-\frac{\left|e\right|}{\lambda }\right), \end{array}$$
where $e\in \left(-\infty ,\infty \right)$ and λ>0 are location and scale parameters, respectively (see, e.g., [15,16]). It was named after Pierre-Simon Laplace (1749–1827), as the distribution whose likelihood is maximized when the location parameter is set to the median. Based on the data (I(1),…,I(T)) the likelihood is equal to
$$\begin{array}{c} \prod_{t=1}^{T}\frac{1}{2\lambda }\exp\left(-\frac{|I\left(t\right)-I_{m}\left(t\right)|}{\lambda }\right). \end{array}$$
It comes that the maximum likelihood estimator of the parameters θ and λ are
$$\begin{array}{c} \left\{\begin{array}{ccc} \hat{\theta } & = & \arg\min\{f_{m}\left(T,\theta \right):\theta \}\\ \hat{\lambda } & = & f_{m}\left(T,\hat{\theta }\right). \end{array}\right. \end{array}$$
In practice, $\hat{\theta }$ are given by an algorithm of optimization, and usually, they are only local minimizers. Having $\hat{\theta }$ and the scale $\hat{\lambda }$, we derive a confidence interval for I(t) with t>T as a solution of the equation
$$\begin{array}{c} \int_{-q}^{q}\frac{1}{2\hat{\lambda }}\exp\left(-\frac{\left|e\right|}{\hat{\lambda }}\right)de=0.95 \end{array}$$
given by $q=-\hat{\lambda }\ln\left(0.05\right)=2.995732\hat{\lambda }$. We derive the confidence interval
$$\begin{array}{c} IC_{0.95}\left[I(t)\right]=\left[{\hat{I}}_{m}\left(t\right)-2.995732\hat{\lambda };{\hat{I}}_{m}\left(t\right)+2.995732\hat{\lambda }\right] \end{array}$$
of I(t) with the confidence level 0.95. Here, ${\hat{I}}_{m}\left(t\right)=\hat{a}\exp\left(-\frac{{\left(t-\hat{l}\right)}^{2}}{{\hat{s}}^{2}}\right)$ in the Gaussian case, and ${\hat{I}}_{m}\left(t\right)=\hat{a}{\hat{z}}^{t}/{\left(\hat{v}+t\right)}^{\hat{s}}$ in the Lerch case.

## 3. LAD Regression Analysis Using Weighted Median

Before going forward, we recall the weighted median definition.

### 3.1. Weighted Median

We recall in the following proposition, the definition and the calculation of the weighted median. For more details, we advise the reader to see the work of [17].

**Proposition** **1.**
*Let us consider a sequence (x(t),w(t)) of real numbers with positive weighted w(t)>0 and t=1,…,T. The minimizer of the function $a\to \sum_{t=1}^{T}w\left(t\right)\left|a-x\left(t\right)\right|$ (called the weighted median) is given as follows. We calculate the permutation p(1),…,p(T), which rearranges the sequence $\left(x(t):t=1,\ldots ,T\right)$ into ascending order. We form the sequence (w(p(t)):t=1,…,T), then we find the largest integer k which satisfies*

$$\begin{array}{c} \sum_{t=1}^{k}w\left(p\left(t\right)\right)\leq \frac{\sum_{t=1}^{T}w\left(t\right)}{2}. \end{array}$$

*If*

$$\begin{array}{c} \sum_{t=1}^{k}w\left(p\left(t\right)\right)<\frac{\sum_{t=1}^{T}w\left(t\right)}{2}, \end{array}$$

*then the weighted median a=x(p(k+1)).*

*If $\sum_{t=1}^{k}w\left(p\left(t\right)\right)=\frac{\sum_{t=1}^{n}w\left(t\right)}{2}$, then the weighted median [x(p(k)),x(p(k+1))] is equal to the interval [x(p(k)),x(p(k+1))].*


#### 3.1.1. Back to Our Proposed LAD Regression

Both our LAD regressions have the form
$$\begin{array}{c} f\left(T,a,b\right)=\frac{1}{T}\sum_{t=1}^{T}\left|ag\left(t,b\right)-I\left(t\right)\right|, \end{array}$$
with a>0, and g:(0,+∞)×D→(0,+∞) is a continuous positive map with *D* is a Euclidean domain. Now, we can announce the following corollary.

**Corollary** **1.**
*For each fixed b, the minima of the function a→f(T,a,b) is attained at the weighted median a(b) of the sequence (x(t)=I(t)/g(t,b):t=1,…,T) endowed with the weights (w(t)=g(t,b):t=1,…,T). Moreover, if $\left(a^{*},b^{*}\right)$ is a local minimizer of the function (a,b)→f(T,a,b), then $a^{*}$ is the weighted median of $\left(x^{*}\left(t\right)=I\left(t\right)/g\left(t,b^{*}\right):t=1,\ldots ,T\right)$ endowed with the weights $\left(w^{*}\left(t\right)=g\left(t,b^{*}\right):t=1,\ldots ,T\right)$.*


**Proof.** We observe that for fixed *b*, the map a→f(T,a,b) is a convex function. Now, let us assume that $\left(a^{*},b^{*}\right)$ is a local minimizer of the function (a,b)→f(T,a,b). Then, $a^{*}$ is the global minimizer of the convex function $a\to f(T,a,b^{*})$. Hence, $a^{*}$ is the weighted median of $\left(x^{*}\left(t\right)=I\left(t\right)/g\left(t,b^{*}\right):t=1,\ldots ,T\right)$ endowed with the weights $\left(w^{*}\left(t\right)=g\left(t,b^{*}\right):t=1,\ldots ,T\right)$.    □

#### 3.1.2. Comparison of the Minimizers of the Map
b→f(T,a(b),b) and the Minimizers of the Map
(a,b)→f(T,a,b)

The following proposition is obvious.

**Proposition** **2.**
*(1) For each fixed a, the map b→f(T,a,b) is above the map b→f(T,a(b),b) and they intersect at the curve a=a(b).*

*(2) If $b^{*}$ is a local minimizer of the map b→f(T,a(b),b), then $\left(a\left(b^{*}\right),b^{*}\right)$ is also a local minimizer of the map (a,b)→f(T,a,b).*

*(3) The local minimizers of the map (a,b)→f(T,a,b) belong to the set {(a(b),b):b}. If $\left(a\left(b^{*}\right),b^{*}\right)$ is a local minimizer of the map (a,b)→f(T,a,b), then, in general, $b^{*}$ is not a local minimizer of the map b→f(T,a(b),b). However, if $\left(a\left(b^{*}\right),b^{*}\right)$ is a global minimizer of (a,b)→f(T,a,b), then $b^{*}$ is also a global minimizer of b→f(T,a(b),b).*


**Proof.** For each fixed *b*, the minima of the map a→f(T,a,b) is attained at the point a(b), which implies that f(T,a,b)≥f(T,a(b),b) and achieves the proof of (1). Let $\left(a^{*},b^{*}\right)$ be a local minimizer of the map (a,b)→f(T,a,b). It follows that $f\left(T,a,b\right)\geq f\left(T,a^{*},b^{*}\right)$ when a∈W,b∈V with W,V are some neighborhoods, respectively, of $a^{*}$ and $b^{*}$. We derive that the minima of $a\in W\to f(T,a,b^{*})$ is attained at $b^{*}$. As $a\in V\to f(T,a,b^{*})$ is convex, it has the unique minima $b^{*}=a\left(b^{*}\right)$. Which achieves the proof of (3). The proof of (2) works as follows. There exists a neighborhood *V* of $b^{*}$ such that
$$\begin{array}{c} f\left(T,a\left(b\right),b\right)\geq f\left(T,a\left(b^{*}\right),b^{*}\right) \end{array}$$
for each b∈V. By definition of a(b), we have f(T,a,b)≥f(T,a(b),b) for all *a*. It follows that $\left(a\left(b^{*}\right),b^{*}\right)$ is a local minimizer of (a,b)→f(T,a,b).    □

**Proposition** **3.**
*Assume that b→f(T,a(b),b) has only a global minimizer. Then, the map (a,b)→f(T,a,b) may have many local minimizers, and b→a(b) is discontinuous at any point $b^{*}$ such that $\left(a\left(b^{*}\right),b^{*}\right)$ is a local minimizer of the map (a,b)→f(T,a,b).*


**Proof.** By definition of the local minimizer, there exists a neighborhood *V* of $\left(a\left(b^{*}\right),b^{*}\right)$ such that $f\left(T,a,b\right)\geq f(T,a\left(b^{*}\right),b^{*})$ for each point (a,b)∈V. Necessarily, (a(b),b) is not in *V* for at least one point *b* near $b^{*}$, if not $f\left(T,a\left(b\right),b\right)\geq f\left(T,a\left(b^{*}\right),b^{*}\right)$ for all point *b* near $b^{*}$, and then $b^{*}$ is a local minimizer of the map b→f(T,a(b),b). This is absurd because b→f(T,a(b),b) has only a global minimizer.    □

## 4. Numerical Results

In China, COVID-19 appeared on 23 December 2019 in the Wuhan region and after its fast-initial spreading, strict rules of social distancing were imposed almost one month later. Three months after the initially reported cases, the spreading in China subsided. The China data in Figure 1 were extracted from owid/COVID-19-data, available on the web.

Figure 1 shows that the peak and location, which are equal, respectively, to *a* = 15,136, and l=22.

### 4.1. LAD Regression Using Gaussian Model with T=10

By varying the initial condition and using the Algorithm 1 with the objectif function equals $\left(l,s\right)\to f_{\mathrm{Gauss}}\left(T,a\left(T,l,s\right),l,s\right)$, we obtain only one minima: $a(10,l^{*},s^{*})=2088.911$, $l^{*}=10.11930$, $s^{*}=5.712179$ and $f_{\mathrm{Gauss}}\left(T,a\left(10,l^{*},s^{*}\right),l^{*},s^{*}\right)=319.5446$. We recall that a(T,l,s) is the weighted median of the sequence $\left(x\left(t\right)=I\left(t\right)\exp\left(\frac{{\left(t-l\right)}^{2}}{s^{2}}\right):t=1,\ldots ,T\right)$ with the weights $\left(w\left(t\right)=\exp\left(-\frac{{\left(t-l\right)}^{2}}{s^{2}}\right):t=1,\ldots ,T\right)$.

However, by varying the initial condition and using the Algorithm 1 with the objectif function equals $\left(a,l,s\right)\to f_{\mathrm{Gauss}}\left(T,a,l,s\right)$, we found several lists of minima. Table 1 shows some of them. The minima 331.487 corresponds to the minimizer $a^{*}=5526.386$, $l^{*}=22.018$, $s^{*}=12.182$. We recall that the observed location is l=22. The minima 337.0095 corresponds to the minimizer $a^{*}$ = 15,185.797, $l^{*}=31.994$, s=15.614. We recall that the observed peak is *a* = 15,136.
**Algorithm 1** The output of the iteration of optim function until convergence**Input:** Fonction *F* 1: initialization $\theta_{0}$, 2: **while** $F\left(\theta_{0}\right)\neq F(T,\mathrm{optim}\left(\theta_{0},F,method=\text{”}Nelder-Mead”\right)$ **do** 3:   $\theta (\theta_{0})=$
optim$\left(\theta_{0},F,method=\text{”}Nelder-Mead”\right)$ // *optim function applied to F with the initialization* $\theta (\theta_{0})$. 4: **end while****output:** $\theta (\theta_{0})$ and $F(\theta \left(\theta_{0}\right))$.


### 4.2. Confidence Intervals Using Six minimas with T = 10 in the Gaussian Model

We recall that the confidence interval of I(t) from the minimizer $\left(\hat{a},\hat{l},\hat{s}\right)$ is given by
$$\begin{array}{c} IC_{0.95}\left[I(t)\right]=\left[\hat{a}\exp\left(-\frac{{\left(t-\hat{l}\right)}^{2}}{{\left(\hat{s}\right)}^{2}}\right)-2.995732\hat{\lambda };\hat{a}\exp\left(-\frac{{\left(t-\hat{l}\right)}^{2}}{{\left(\hat{s}\right)}^{2}}\right)+2.995732\hat{\lambda }\right]\;\;\;\;\mathrm{with}\;\;\;\;t>10. \end{array}$$

In Figure 2, we present the confidence intervals for the global minima and five local minimas among the list T=10. The predictions using the minimas 335.047 and 336.92 are clearly bad. However, their predictions at the location l=22 are close to the real peak among the six minimas. The percentage of predictions is given in Table 1. An R source code is given in Appendix A, which can be used to determine the confidence intervals for the other values of *T* once the list of minimas is determined by using one of the three considered methods.

In Figure 2, we present the confidence intervals for the global minima and six local minimas among the list T=10 in the Gaussian model.

**Remark** **1.***Let us conclude this paragraph with a comparison between the output of the* *optim* *function and the output of the iteration of the* *optim* *function until the convergence using the same initialization with T=10. As initialization, we use (a=25,325.01, l=41.78141, s=19.85630) for the two approaches; the use of the* *optim* *function leads to the following minima (23,034.33,36.03957,16.80575,338.2104), while the iteration of the* *optim* *function until convergence leads to (2137.277,10.98621,6.457341,320.9898). Figure 3 illustrates the plots of daily infections t∈[1,60]→I(t) of COVID-19 in China, the LAD regression $t\in \left[1,60\right]\to I_{without}^{*}\left(t\right)$ using the minima 338.2104 of the* *optim* *function and the LAD regression $t\in \left[1,60\right]\to I_{iteration}^{*}\left(t\right)$ using the minima 320.9898 of the iteration of the* *optim* *function until the convergence.*

### 4.3. LAD Regression Using the Lerch Model with T=10

By varying the initial condition and using the Algorithm 1 with the function $\left(z,v,s\right)\to f_{Lerch}\left(T,a\left(T,z,v,s\right),z,v,s\right)$ we found a huge number of minimas. Table 2 shows some of them. We recall that a(T,z,v,s) is the weighted median of the sequence $(x\left(t\right)=I\left(t\right){\left(v+t\right)}^{s}/z^{t}$ : t=1,…,T) with the weights $\left(w\left(t\right)=z^{t}/{\left(v+t\right)}^{s}:t=1,\ldots ,T\right)$. We also recall that in the Gaussian case, the surface $\left(l,s\right)\to f_{\mathrm{Gauss}}\left(T,a\left(T,l,s\right),l,s\right)$ has only one minima equal to the global minima of the map $\left(a,l,s\right)\to f_{\mathrm{Gauss}}\left(T,a,l,s\right)$.

Figure 4 and Figure 5 show, respectively, the optimal time series for the Lerch and Gauss models for T=10, using our list of local minimas. We can observe that the Lerch model fits better with the prediction of I(t) for t>10. But the Gauss model fits better with the prediction of the location and the size of the peak.

In Table 3, we report the columns of mode positions and the corresponding amplitudes for each minimizer.

### 4.4. Confidence Intervals Using Six Minimas with T=10 in the Lerch Model

The percentage of predictions is given in Table 2. Observe that the best percentage of predictions occurs at the global minima of the Lerch model, but the best percentage of predictions occurs at the local minimizer 325.186 of the Gauss model.

In Figure 6, we present the confidence intervals for the global minima and six local minimas among the list T=10 of the Lerch model.

## 5. The Case *T* > 10

In this section, we consider the cases T=20 and T=60. In the Lerch model case, LAD regression still has many minimizers for T=20. See Table 4. But in the Gauss model, LAD regression has only one minimizer for T=20. See Table 5.

From Table 4 and Table 5, we can observe that the percentage of prediction of the Lerch model is better than the percentage of prediction of the Gauss model.

Finally, the minima for T=60 for both models (Gauss and Lerch) are unique and respective to (3318.433, 16.94084, 10.19735, 617.2386) and (0.05691053, 0.7445283, 3.3413322, −5.2886448, 570.7861098).

Figure 7 shows the optimal time series for the Lerch and Gauss models for T=60, using our local minima.

We observe that the Lerch approximation has a heavier tail than the Gaussian approximation.

## 6. Conclusions

In this work, we considered LAD nonlinear regression
$$\begin{array}{c} f\left(T,\theta \right)=\frac{\sum_{t=1}^{T}\left|I\left(t\right)-I_{m}\left(t\right)\right|}{T}, \end{array}$$
with the subscript m=Gauss (respectively, m=Lerch).

The two models have the form $\left(a,b\right)\to f\left(T,a,b\right)=\sum_{t=1}^{T}\left|ag(t,b)-I(t)\right|$. We have associated f(T,·) with the map S(T,·):b→f(T,med(x(b),w(b)),b) with the sequence x(b)=(I(t)/f(t,b),t=1,…,T), and the weights w(b)=(g(t,b),t=1,…,T), and med(x(b),w(b)) denotes the weighted median of x(b) endowed with the weights w(b). We showed that if $b^{*}$ is a local minimizer of *S*, then $\left(med\left(x\left(b^{*}\right),w\left(b^{*}\right)\right),b^{*}\right)$ is also a local minimizer of f(T,·). The converse is, in general, false, i.e., if $\left(a^{*},b^{*}\right)$ is a local minimizer of f(T,·), then $b^{*}$ is not, in general, a local minimizer of *S*. However, if $\left(a^{*},b^{*}\right)$ is the global minima of f(T,·), then $b^{*}$ is also the global minima of *S*. We showed that if *S* has only a global minima, then b→f(T,med(x(b),w(b)),b) is discontinuous at each local minimizer. Using the data of the daily infections of COVID-19 in China during the first wave, we showed numerically in the Gaussian case $g\left(t,l,s\right)=\exp\left(-\frac{{\left(t-l\right)}^{2}}{s^{2}}\right)$ that the map f(T,a,l,s) has a huge number of local minimas, but the surface *S* has only a global minima, which is also the global minima of the map f(T,·). However, in the Lerch case $g\left(t,z,v,s\right)=\frac{z^{t}}{{\left(t+v\right)}^{s}}$, contrary to the Gaussian case, we showed that the maps (a,z,v,s)→f(T,a,z,v,s) and (z,v,s)→S(T,z,v,s) have each a huge number of local minimas. We derived confident intervals for the daily infections from each local minima. Our message is that each local minima contains a part of the information and can be used for the prediction of a part of the parameters.

In future work, we will deal with the log-transformation $\ln\left(I_{Gauss}\left(t\right)\right)=\beta_{0}+\beta_{1}t-\beta_{2}t^{2}$, where $\left(\beta_{0},\beta_{1},\beta_{2}\right)$ are in bijection with the parameters *a*, *l*, *s* of the Gaussian model. This allows us to consider the linear model $\ln\left(I\left(t\right)\right)=\beta_{0}+\beta_{1}t-\beta_{2}t^{2}+\varepsilon \left(t\right)$, with the errors (ε(t)), which are i.i.d. We will deal with this problem using other robust regression estimators as the least trimmed median estimator (LTM) proposed in [18].

## Figures and Tables

**Figure 1 entropy-25-01474-f001:**
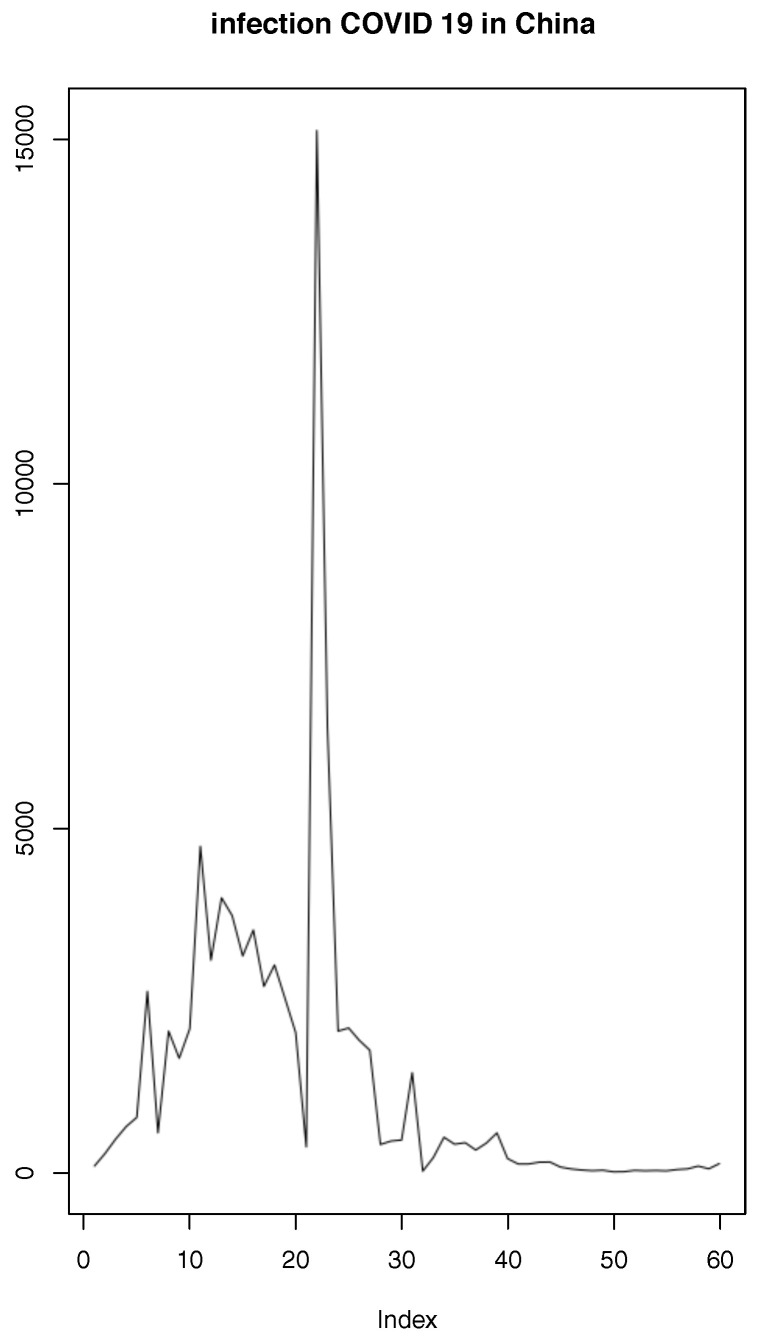
Daily infections of COVID-19 in China.

**Figure 2 entropy-25-01474-f002:**
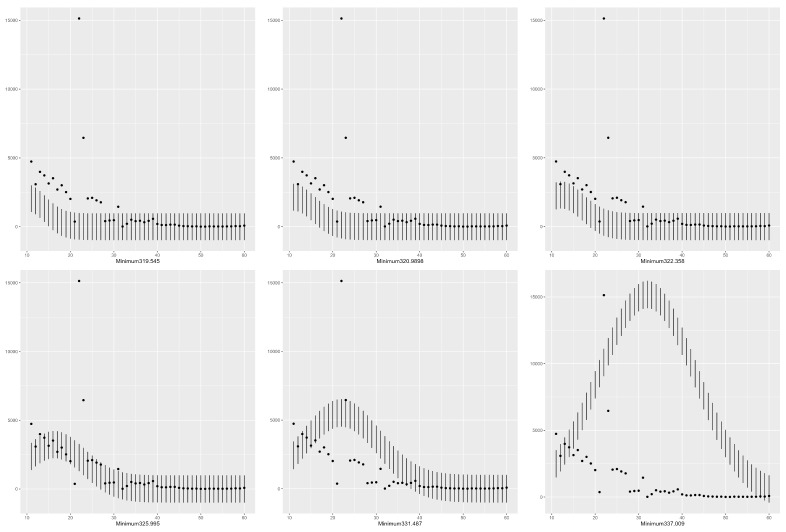
Confidence intervals using six minimas with T = 10 in the Gaussian model.

**Figure 3 entropy-25-01474-f003:**
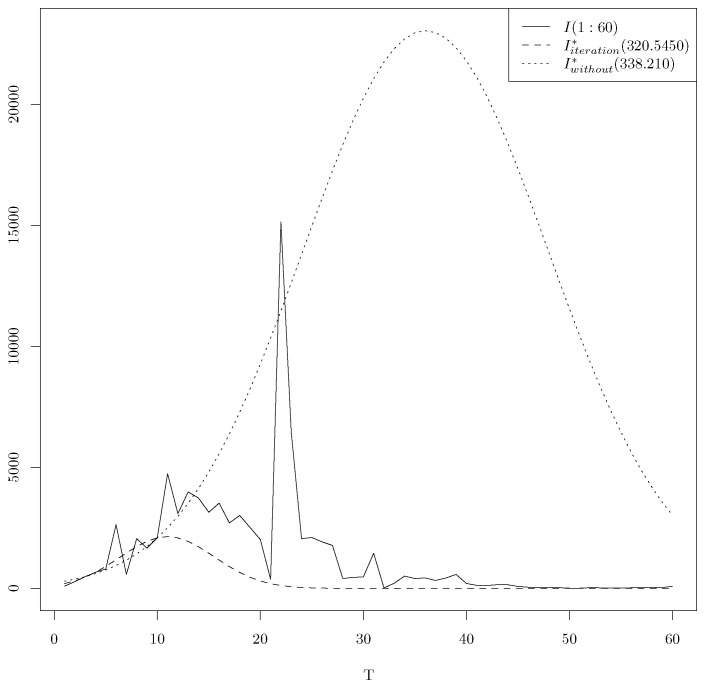
Optimal time series for T=10, for the Gaussian model with and without iterations.

**Figure 4 entropy-25-01474-f004:**
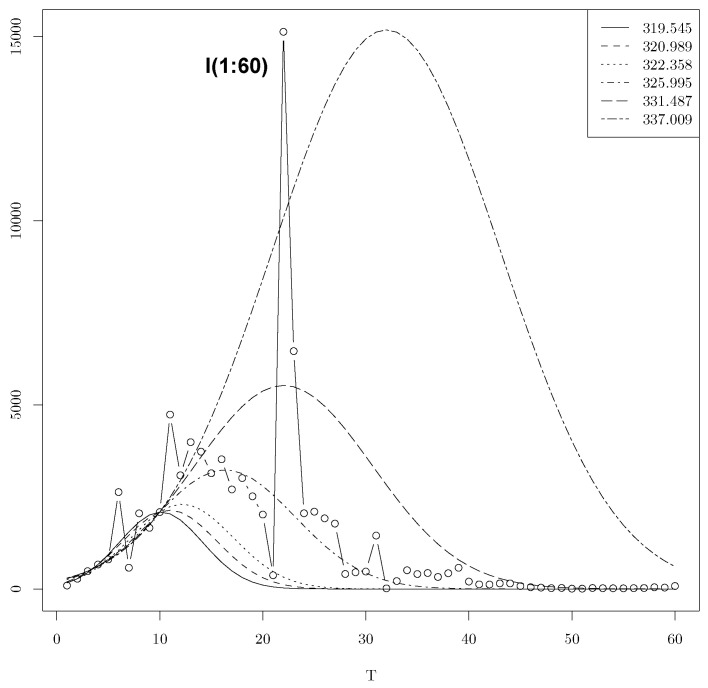
Optimal time series for T=10 for Gauss.

**Figure 5 entropy-25-01474-f005:**
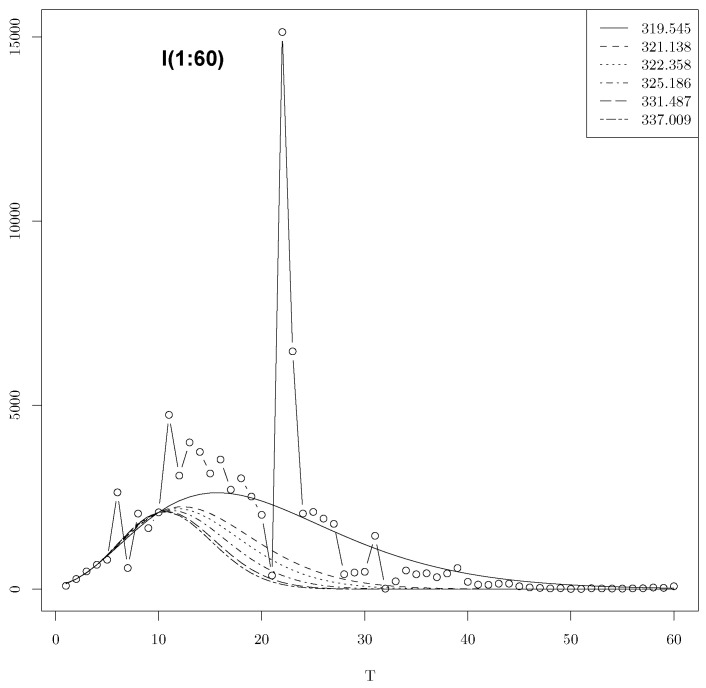
Optimal time series for T=10 for Lerch.

**Figure 6 entropy-25-01474-f006:**
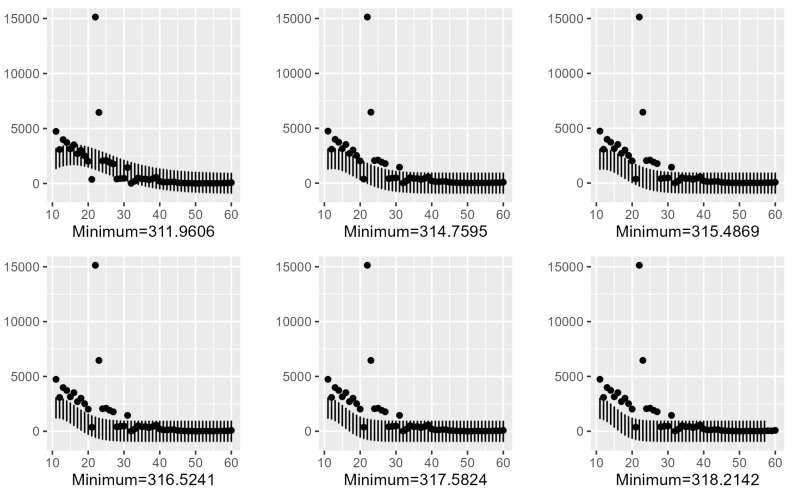
Confidence intervals using six minimas with *T* = 10 of the Lerch model.

**Figure 7 entropy-25-01474-f007:**
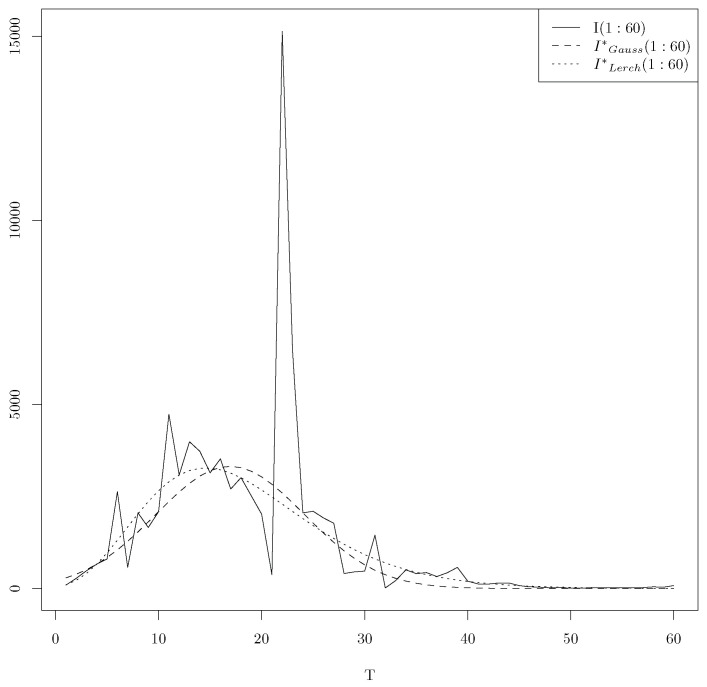
Optimal time series for T=60 for Lerch and Gauss.

**Table 1 entropy-25-01474-t001:** A list of minimizers of $\left(a,l,s\right)\to f_{\mathrm{Gauss}}\left(T,a,l,s\right)$ with T=10.

$a^{*}$	$l^{*}$	$s^{*}$	minima	% Predictions
2088.911	10.119	5.712	319.545	0.66
2137.277	10.98621	6.457	320.9898	0.66
2301.818	12.186	7	322.358	0.68
3229.843	16.346	9.609	325.995	0.88
5526.387	22.018	12.183	331.487	0.60
15,185.797	31.994	15.614	337.009	0.12

**Table 2 entropy-25-01474-t002:** A list of minimizers of $\left(a,z,v,s\right)\to f_{\mathrm{Lerch}}\left(T,a,z,v,s\right)$ with T=10.

$a^{*}$	$z^{*}$	$v^{*}$	$s^{*}$	Minima	% Predictions
5.990886 $\times 10^{-1}$	0.81023163	3.201870	−3.976447	311.9606	0.84
1.354215 $\times 10^{-8}$	0.59318684	8.174956	−10.683932	314.7595	0.70
1.555525 $\times 10^{-14}$	0.50711308	10.701817	−15.255833	315.4869	0.68
8.235320 $\times 10^{-34}$	0.35420137	16.512305	−28.741171	316.5241	0.66
2.185815 $\times 10^{-103}$	0.1647315	28.956740	−71.553876	317.5824	0.66
7.23785 $\times 10^{-266}$	0.05804243	45.948388	−160.673224	318.2142	0.66

**Table 3 entropy-25-01474-t003:** Mode position and amplitude using Lerch.

$a^{*}$	$z^{*}$	$v^{*}$	$s^{*}$	Mode Position	Amplitude
5.990886 $\times 10^{-1}$	0.81023163	3.201870	−3.976447	16.19885	2618.210
1.354215 $\times 10^{-8}$	0.59318684	8.174956	−10.683932	12.78678	2236.537
1.555525 $\times 10^{-14}$	0.50711308	10.701817	−15.255833	12.26928	2186.184
8.235320 $\times 10^{-34}$	0.35420137	16.512305	−28.741171	11.68264	2134.780
2.185815 $\times 10^{-103}$	0.1647315	28.956740	−71.553876	11.22172	2101.053
7.23785 $\times 10^{-266}$	0.05804243	45.948388	−160.673224	11.05461	2090.932

**Table 4 entropy-25-01474-t004:** A list of minimizers of $\left(a,z,v,s\right)\to f_{\mathrm{Lerch}}\left(T,a,z,v,s\right)$ with T=20. The last column contains the percentage of predictions.

$a^{*}$	$z^{*}$	$l^{*}$	$s^{*}$	Minima	% Predictions
4.366541 × 10^−268^	0.1073967	55.31169	−154.51447	392.5692	0.825
3.422419 × 10^−262^	0.1387492	47.81770	−121.91023	393.5286	0.825
2.572282 × 10^−256^	0.1708497	41.39999	−97.69044	394.7033	0.825
1.155039 × 10^−253^	0.1860852	38.76658	−88.52397	395.2740	0.825
3.035334 × 10^−234^	0.3069971	23.34634	−43.86187	400.4082	0.825
4.970115 × 10^−230^	0.3344249	20.71433	−37.77693	401.7952	0.825
1.667726 × 10^−233^	0.3634773	22.30914	−36.60867	403.6049	0.825

**Table 5 entropy-25-01474-t005:** The unique minimizer of $\left(a,l,s\right)\to f_{\mathrm{Gauss}}\left(T,a,l,s\right)$ with T=20.

$a^{*}$	$l^{*}$	$s^{*}$	Minima	% Predictions
3712.1297	14.2556	7.60744	386.0318	0.80

## Data Availability

Data are unavailable due to privacy.

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
