# Peer review of "Gaussian and Lerch Models for Unimodal Time Series Forcasting"

_entropy, 2023, doi:10.3390/e25101474_

Round 1
Reviewer 1 Report
The manuscript focuses on linear regression with LAD instead of sum of squares.
However, the manuscript is hard to read. For example, the abstract contains a lot of undefined notations that confused me a lot. Like what are a, l ,s, x?
I also don't think the paper is really ready to be published. For example, what's the reason to make Figure 1-2 that big? Is Figure 3 a screenshot from R? This is very unprofessional I believe.
I strongly recommend the authors to polish the paper carefully before resubmitting it to any venue.
See above comments.
Author Response
Reviewer 1:
However, the manuscript is hard to read. For example, the abstract contains a lot of undefined notations that confused me a lot. Like what are a, l, s, x? I also don’t think the paper is really ready to be published. For example, what’s the reason to make Figure 1-2 that big? Is Figure 3 a screenshot from R? This is very unprofessional I believe.
I strongly recommend the authors to polish the paper carefully before resubmitting it to any venue.
The authors’s answer :
We are very grateful to your comments and thoughtful suggestions. In the revised version of the current version, we made many change according to your comment.

Reviewer 2 Report
The purpose of the paper is to introduce methods which can help in finding the global minimum in the objective function specified by the least absolute deviation (LAD).
In my opinion the paper provides a series of interesting ideas. The proofs seems correct to me. Here is a list of suggestions that could improve the paper.
11) Nowadays LAD is seldomly used due to fact that this technique has a 0 breakdown point. More specifically the use of LMS, LTS, S, MM or the forward search are the standard techniques in robust regression analysis. At least the authors should refer to the above methods in the paper. These methods are based on subsets in the sense that given a subset of size p (where p is the number explanatory variables) the objective function is computed for all the n observations using parameter estimates based on subset. After extracting (say) 10000 subsets and for each one we take the smallest objective function. It is not clear to me how the suggested procedure compares with the standard one described above as function of n (sample size) and p (also for computational time)
22) Probably the time series data (which contain autocorrelation) are not the best possible example. It was certainly better to use iid data.
33) It would be nice if the authors could bring an example where the application of standard LAD using pure optim fails and on the other hand their method succeeds.
44) Appendix G can be removed because it contains things which can be found in any book of robust statistics.
The quality of the english seems fine to me
Author Response
Reviewer 2:
The purpose of the paper is to introduce methods which can help in finding the global minimum in the objective function specified by the least absolute deviation (LAD). In my opinion the paper provides a series of interesting ideas. The proofs seems correct to me. Here is a list of suggestions that could improve the paper.
Nowadays LAD is seldomly used due to fact that this technique has a 0 breakdown point. More specifically the use of LMS, LTS, S, MM or the forward search are the standard techniques
in robust regression analysis. At least the authors should refer to the above methods in the paper. These methods are based on subsets in the sense that given a subset of size p (where p is the number explanatory variables) the objective function is computed for all the n observations using parameter estimates based on subset. After extracting (say) 10000 subsets and for each one we take the smallest objective function. It is not clear to me how the suggested procedure compares with the standard one described above as function of n (sample size) and p (also for computational time)
The authors’s answer :
We are very grateful to your comments and thoughtful suggestions. It is a very interesting addition of the paper, based on your suggestion, we made many change in the paper. We compare further work the use of non-linear regression model based on non-gaussian densities, such as Lerch one (Contreras-Reyes, 2021).
We also added the following three references:
Johnson, N.L., Kotz S., Balakrishnan, N. (1994). Continuous Univariate Distributions, Wiley.
Contreras-Reyes, J.E. (2021). Lerch distribution based on maximum nonsymmetric entropy principle: Application to Conway’s game of life cellular automaton. Chaos, Solitons & Fractals, 151, 111272.
Christophe Crous, Peter J. Rousseeuw, and Annemie Van Bael, (1996). Positive-breakdown regression by minimizing nested scale estimators, J. Statist. Plann. Inference, Volume 53, Issue 2, 15 August 1996, Pages 197–235
Reviewer 2:
Probably the time series data (which contain autocorrelation) are not the best possible example. It was certainly better to use iid data.
The authors’s answer :
Thank you very much for pointing out this problem. Now in the revised version of the paper, we added this paragraph in the conclusion of the paper:
In a future work we will deal with the log-transformation ln(I Gauss (t)) = β 0 + β 1 t − β 2 t 2 , where ( β 0 , β 1 , β 2 ) are in bijection with the parameters a, l, s of the Gaussian model. This allows us to consider the linear model ln(I(t)) = β 0 + β 1 t − β 2 t 2 + ε(t), with the errors (ε(t)) are i.i.d. We will deal with this problem using others robust regression estimators as the least trimmed median estimator (LTM) proposed in Crous et al. (1996).
Reviewer 2:
It would be nice if the authors could bring an example where the application of standard LAD using pure optim fails and on the other hand their method succeeds.
The authors’s answer :
Thank you very much for your suggestion.
In the revised version of this paper, we added a comparison between the proposed iterative approach and the standard optim.
Reviewer 2:
Appendix G can be removed because it contains things which can be found in any book of robust statistics.
The authors’s answer :
We are very grateful to your comments and thoughtful suggestions. Now, in the revised paper, we removed Appendix G.

Reviewer 3 Report
Review of "Confidence intervals from local minimums of objective function" by Dermoune et al. (2023)
Authors developed an interesting paper related to optimization methods for non-linear regression modelling. They descrive the optimization methods, their properties and derive confidence intervals of estimators. Methods are applied to China's COVID-19 data.
I think that paper is suitable for Entropy journal; however, some issues must be addressed by the authors before consider publication in the journal:
1. End of page 1: x_i is not 0, for all i.
2. Page 2: some "))" are missing for "\theta_2(k))".
3. Page 3: the first equation looks like the Laplace distribution (Johnson et al., 1994), if yes, call it.
4. Page 5: in Proof 4.5 of Proposition 4.4., it does not appear 1-3.
5. Page 5: "Necessarly" <-> "Necessarily".
6. Page 5, step 2: Why a truncated Gaussian, and truncated in what interval? Perhaps authors could reference Appendix G. See also Step of Algorithm 2. 7. Page 6: "of China (see Figure 1) are extracted from owid/covid-19-data available on the web."
8. Page 7: How was determined these initial parameters? visual inspection?
9. Page 7: revise sentence "We observe the convergence of optim function in 2". It is unclear.
10. Page 8: why are produced these large bounds?
11. Page 8: Results 3 are not an adequate form to present these results. Use a figure: index (1,...,100) vs. value. Also for Table 3.
12. Figure 4: why these points are out from IC?
13. Page 11: in Conclusions, These equations are repeated from body's manuscript. These sentences could only describe the method.
14. Page 12: at the end of Conclusions section, far from assuming a Gaussian model, authors could add as a further work the use of non-linear regression model based on non-gaussian densities, such as Lerch one (Contreras-Reyes, 2021).
15. Appendix H.1 opt() function: the parameter function: method = "Nelder-Mead", could be added to optim to ensure that these method was used.
16. Page 13: Reference title is missing.
References:
Johnson, N.L., Kotz S., Balakrishnan, N. (1994). Continuous Univariate Distributions, Wiley.
Contreras-Reyes, J.E. (2021). Lerch distribution based on maximum nonsymmetric entropy principle: Application to Conway’s game of life cellular automaton. Chaos, Solitons & Fractals, 151, 111272.
I have detected only one grammar error in Page 5: "Necessarly" <-> "Necessarily".
Author Response
Reviewer 3:
Authors developed an interesting paper related to optimization methods for non-linear regression modelling. They descrive the optimization methods, their properties and derive confidence intervals of estimators. Methods are applied to China’s COVID-19 data. I think that paper is suitable for Entropy journal; however, some issues must be addressed by the authors before consider publication in the journal: 1. End of page 1: x i is not 0, for all i.
2. Page 2: some "))" are missing for " θ 2 (k))".
5. Page 5: "Necessarly" <-> "Necessarily".
The authors’s answer :
Thank you very much for pointing out this omission, we correct this mistakes.
Reviewer 3:
3. Page 3: the first equation looks like the Laplace distribution (Johnson et al., 1994), if yes, call it.
The authors’s answer :
We are very grateful to your comments and thoughtful suggestions. In the revised version of the paper, we added the following sentences in the second section.
where e ∈ (−∞, ∞) and λ > 0 are location and scale parameters, respectively (see, e.g., Ord (1983), Johnson et al. (1994)). It was named after Pierre-Simon Laplace (1749-1827), as the distribution whose likelihood is maximized when the location parameter is set to the median.
Reviewer 3:
4. Page 5: in Proof 4.5 of Proposition 4.4., it does not appear 1-3.
The authors’s answer :
Thank you for your comment, in the revised version, we have added the following developement :
For each b fixed the minimum of the map a → f(T, a, b) is attained at the point a(b), which implies that f(T, a, b) ≥ f(T, a(b), b) and achieves the proof of 1). Let (a ∗ , b ∗ ) be a local minimizer of the map (a, b) → f(T, a, b). It follows that f(T, a, b) ≥ f(T, a ∗ , b ∗ ) when a ∈ W, b ∈ V with W, V are some neighborhoods respectively of a ∗ and b ∗ . We derive that the minimum of a ∈ W → f(T, a, b ∗ ) is attained at b ∗ . As a ∈ V → f(T, a, b ∗ ) is convex, it has the unique minimum b ∗ = a(b ∗ ). Which achieves the proof of 3).
Reviewer 3:
6. Page 5, step 2: Why a truncated Gaussian, and truncated in what interval? Perhaps authors could reference Appendix G. See also Step of Algorithm 2. 7. Page 6: "of China (see Figure 1) are extracted from owid/covid-19-data available on the web."
The authors’s answer :
Thank you very much for your nice comments, now, in the revised version according to your comments and the two others reviewers, we made our best to give all the needed informations for best understanding of the proposed approach.
Reviewer 3:
8. Page 7: How was determined these initial parameters? visual inspection?
The authors’s answer :
Thank you very much for your nice comment, in the revised version of this paper, we added many visual inspections and we precise the initial parameters.
Reviewer 3:
9. Page 7: revise sentence "We observe the convergence of optim function in 2". It is unclear.
The authors’s answer :
Thank for your comment, in the revised version of this paper, we revised this part of the paper.
Reviewer 3:
10. Page 8: why are produced these large bounds?
The authors’s answer :
Thank your for your kind comment, in the revised version of this paper, we have added some informations on our estimation, pourcentage of prediction, mode position and amplitude, see Tables 1, 2, 3, 4 and 5.
Reviewer 3:
11. Page 8: Results 3 are not an adequate form to present these results. Use a figure: index (1,...,100) vs. value. Also for Table 3.
The authors’s answer :
Thank you very much for your comments, in the revised version we added a visual perception.
Reviewer 3:
12. Figure 4: why these points are out from IC?
The authors’s answer : Thank you very much for your comment, in the revised version of this paper, we report the pourcentage of prediction using the two methods.
Reviewer 3:
13. Page 11: in Conclusions, These equations are repeated from body’s manuscript. These sentences could only describe the method.
The authors’s answer : Thank you very much for your comment, in the revised version of this paper, we made many change in the conclusion according to your comment.
Reviewer 3:
14. Page 12: at the end of Conclusions section, far from assuming a Gaussian model, authors could add as a further work the use of non-linear regression model based on non-gaussian densities, such as Lerch one (Contreras-Reyes, 2021).
The authors’s answer :
Thank you very much for this nice proposition, it was an addition of our paper, based on your comment, we made many change in this paper, firstly, we change the title of the paper, and we focus on both Gaussian and Lerch models for unimodal time series forcasting.
Reviewer 3:
15. Appendix H.1 opt() function: the parameter function: method = "Nelder-Mead", could be added to optim to ensure that these method was used.
The authors’s answer :
Thank you very much for your nice comment, we correct this in the revised version of this paper.
Reviewer 3:
16. Page 13: Reference title is missing.
The authors’s answer :
Based on your comment, we added the following three references:
Johnson, N.L., Kotz S., Balakrishnan, N. (1994). Continuous Univariate Distributions, Wiley.
Contreras-Reyes, J.E. (2021). Lerch distribution based on maximum nonsymmetric entropy principle: Application to Conway’s game of life cellular automaton. Chaos, Solitons & Fractals, 151, 111272.
Christophe Crous, Peter J. Rousseeuw, and Annemie Van Bael, (1996). Positive-breakdown regression by minimizing nested scale estimators, J. Statist. Plann. Inference, Volume 53, Issue 2, 15 August 1996, Pages 197–235
Again, we appreciate all your insightful comments. Thank you for taking the time and energy to help us improve the paper.

Round 2
Reviewer 2 Report
At the end of the paper I would add the following sentence
In a future work we will also deal with a comparison of the currennt estimators with those based on MM (Yohai et al. 2006) and Forward Search estimators (Atkinson et al. 2004).
Nothing to point out
Reviewer 3 Report
In thi new version, it is rewarding to see how Lerch model help to authors to increase the value of the manuscript, giving new results compared with Gaussian one. In my opinion, the paper could be considered for publication in Entropy journal. I don't have further comments.
Some minor issues could be addressed; for example, in the page 11, about the use of commas, "and", etc., in: "However, in Lerch case g(t, z, v, s) and in contrary to Gaussian case, we showed that the maps". I recommend to authors a second review of grammar and typos errors.